# HUMAN-LIKE GEOMETRIC ABSTRACTION IN LARGE PRE-TRAINED NEURAL NETWORKS

**Declan Campbell** [1*] **, Sreejan Kumar** [1,2*] **, Tyler Giallanza** [3] **, Thomas L. Griffiths** [3,4] **, Jonathan D. Cohen** [1, 3]

[1] Princeton Neuroscience Institute, [2] NYU Center for Data Science,
[3] Princeton University Dept of Psychology, [4] Princeton University Dept of Computer Science

## ABSTRACT

Humans possess a remarkable capacity to recognize and manipulate abstract structure, which is especially apparent in the domain of geometry. Recent research in cognitive science suggests neural networks do not share this capacity, instead proposing that human geometric reasoning abilities come from discrete symbolic structure in human mental representations. However, progress in artificial intelligence (AI) suggests that neural networks begin to demonstrate more human-like reasoning after scaling up standard architectures in both model size and amount of training data. In this study, we revisit empirical results in cognitive science on geometric visual processing and identify three key biases in geometric visual processing: a sensitivity towards complexity, a preference for regularity, and the perception of parts and relations. We test tasks from the literature that probe these biases in humans and find that large pre-trained neural network models used in modern forms of AI demonstrate human-like biases in abstract geometric processing.

## 1 INTRODUCTION

Humans have an amazing capability to build useful abstractions that can capture regularities in the external world. By forming abstractions that can generalize to future experience, humans are able to exhibit efficient learning and strong generalization across domains (Lake et al., 2015; Hull, 1920). One domain in which this has been observed by cognitive scientists is geometric reasoning (Dehaene et al., 2022), where people consistently extract abstract concepts, such as parallelism, symmetry, and convexity, that generalize across many visual instances.

These observations, together with rigorous empirical work examining visual perception of geometric forms (Sablé-Meyer et al., 2021; 2022) have led some cognitive scientists to hypothesize that human abstractions arise from symbols that are used recursively and compositionally to produce abstractions necessary for geometric reasoning. Such hypotheses recapitulate the "Language of Thought" hypothesis of Fodor 1975: that higher-order cognition in humans is the product of recursive combinations of pre-existing, conceptual primitives, analogous to the way in which sentences in a language are constructed from simpler elements, such as words and phrases. It has sometimes been assumed that, as a consequence of this hypothesis, artificial neural networks cannot produce such abstractions without the exogenous addition of symbolic primitives and/or processing machinery (Fodor & Pylyshyn, 1988; Marcus, 2018). Indeed, empirical work in this domain has shown that explicitly symbolic models fit human behavior better than standard neural networks (Sablé-Meyer et al., 2021; Bowers et al., 2023) without such additions.

A recent paradigm shift in the field of artificial intelligence has begun to offer a challenge to the LoT hypothesis. Large neural networks, trained on massive datasets, are starting to demonstrate reasoning abilities similar to those of humans in linguistic and analogical reasoning tasks (Bubeck et al., 2023; Webb et al., 2023b; Wei et al., 2022). These models rely on continuous vector space representations rather than discrete symbols, and lack any explicit symbolic machinery (McCoy et al., 2018).

---

* equally contributing co-first authors

In this article, we test whether such large pre-trained neural network models demonstrate a similar capacity for abstract reasoning in the domain of geometry. Specifically, we apply neural network models to behavioral tasks from recent empirical work (Sablé-Meyer et al., 2021; 2022; Hsu et al., 2022) that catalogue three effects indicative of abstraction in human geometric reasoning. First, humans are sensitive to *geometric complexity*, such that they are slower to recall complex images as compared to simpler ones (Sablé-Meyer et al., 2022). Second, humans are sensitive to *geometric regularity* (based on features such as right angles, parallel sides, and symmetry) such that they are able to classify regular shapes (such as squares) more easily than less regular ones (such as trapezoids) (Sablé-Meyer et al., 2021). Third, humans decompose geometric objects into *geometric parts and relations* when learning new geometric categories, and generalize these to unseen stimuli (Hsu et al., 2022).

We apply existing neural network models trained on large databases of images and text to three tasks corresponding to each of these effects (Fig. 1). We then evaluate how well the models reproduce human behavior and examine the extent to which the models' internal representations support abstract geometric reasoning. The results demonstrate that large neural networks, when trained on sufficiently rich data, can in some situations show preferences for abstraction similar to those observed in humans, providing a connectionist alternative to symbolic models of geometric reasoning.

## 2 NEURAL NETWORK MODELS

To test our hypothesis, we used two self-supervised transformer-based neural network models (DINOv2 and CLIP) that had been pre-trained on massive datasets of images and text. DINOv2 is a large Vision Transformer (Dosovitskiy et al., 2020) with 1B parameters trained on a self-supervised objective.

**DINOv2** has two main losses — an image level loss, where embeddings of affine augmentations of the same image are trained to have maximal similarity using a student-teacher network configuration (Caron et al., 2021) and a patch-level loss, in which random patches of images are masked out and the network has to fill them in (Zhou et al., 2021).

**CLIP** is a similar transformer-based architecture that is trained on a joint vision-language objective by maximizing the cosine similarity between image embeddings and their corresponding language embeddings.

We also used a standard convolutional neural network (**ResNet-50**; He et al. 2016), which is pre-trained on a basic image classification task on the ImageNet dataset (Deng et al., 2009). Both DINOv2 and CLIP are trained on datasets that are orders of magnitude larger than the ImageNet dataset on which the ResNet model was trained (e.g., DINOv2's dataset LVD-142M contains 142 million examples, whereas ImageNet contains 1.2M examples). We hypothesized that these larger models would have a greater opportunity to discover more general and systematic geometric features as a consequence of the size and scope of the datasets on which they are trained. Thus, they would exhibit more human-like sensitivity to geometric regularities in visual processing tasks than models trained on less data (such as ResNet).

In each section below we consider the performance of these models with respect to each of the three types of systematicity observed for humans in processing geometric figures: geometric complexity, geometric regularity, and geometric parts and relations.

## 3 GEOMETRIC COMPLEXITY

### 3.1 BACKGROUND

Sablé-Meyer et al. 2022 formalized the concept of subjective complexity of geometric shapes using program induction, as implemented in the DreamCoder framework (Ellis et al., 2021). This broadly follows the Language of Thought (LoT) schema (Quilty-Dunn et al., 2023), modeling a geometric shape through a generative drawing program consisting of a set of motor commands that trace an object using a virtual pen. The motor commands come from a range of motor primitives specified by a domain-specific language (such as tracing a curve or changing direction) and combinatory primitives (such as Concat, which concatenates two subprograms; or Repeat, which repeats a subprogram

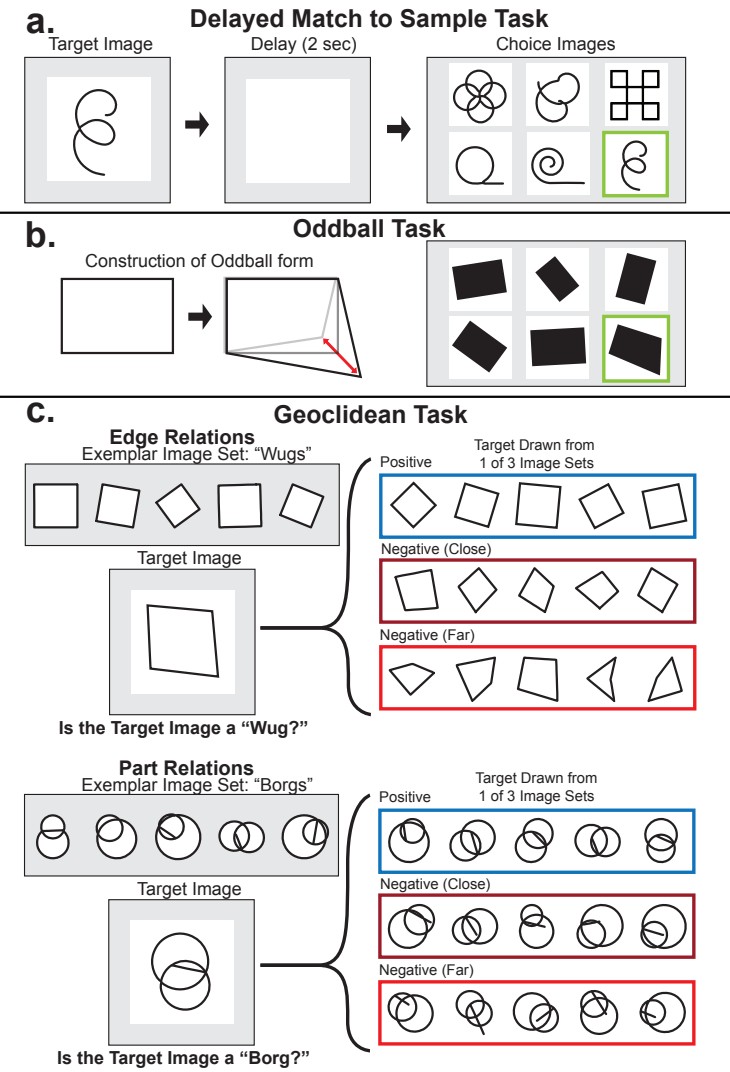

Figure 1: **Tasks.** (A). Delayed-match to sample task involving hierarchically structured shapes generated using the Dreamcoder DSL. (B). Quadrilateral oddball task in which humans and machines are evaluated on their sensitivity to geometric regularity and symmetry. (C). Category judgment task involving geometric figures with hierarchical shapes.

a specified number of times). These symbolic programs are then rendered into images like the ones shown in Fig. 1a. Sablé-Meyer et al. 2022 quantified the complexity of each image using the length of the program (the Minimum Description Length; MDL), that was used to draw it.

To validate MDL as a metric for subjective geometric complexity in humans, Sablé-Meyer et al. 2022 designed a working memory task using stimuli rendered from the above LoT model (Fig. 1a). Participants were given however much time they needed to commit a geometric stimulus to memory before pressing a button to end the memorization phase. After a memorization phase, they were faced with a blank screen for two seconds, followed by the presentation of six stimuli comprised of the the original stimulus they had memorized (the "target image") and five distractors. The task was to identify the target image. Human participants performed well at this task, with error rates as low as 1.82%. Critically, reaction times (RTs) were longer for targets with higher MDLs. (Fig. 2b).

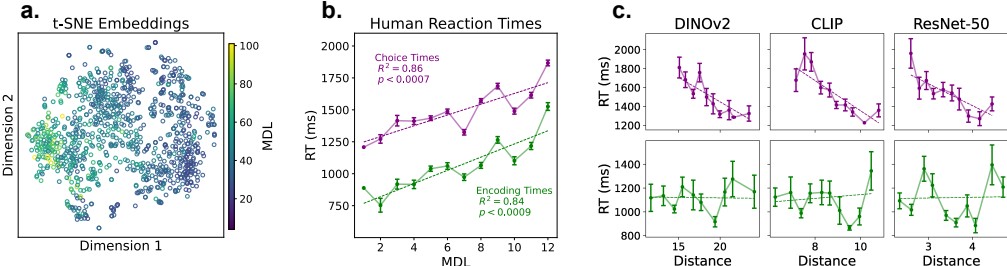

Figure 2: **Model embeddings predict human performance on DMTS task.** a) t-SNE plot of model embeddings of 1000 stimuli generated from randomly sampled programs from the LoT model used in Sable-Meyer et al. (2022), colored according to stimulus MDL. Stimulus embeddings implicitly encode variation in MDL, that was also evaluated with regression analysis. b) Human choice and encoding reaction times on the task plotted with a linear regression between MDL and reaction time. $R^2$ values and $p$-values from fitting GLM model with same confounds as originally used in Sable-Meyer et al. (2022). c) Embedding metrics for all three models plotted against choice (top) and encoding (bottom) RT. $R^2$ values and $p$-values from GLM model with same confounds as originally used in Sable-Meyer et al. (2022).

## 3.2 RESULTS

We tested the extent to which large neural network models (DINOv2 and CLIP), but not the smaller one (ResNet-50) would capture these features, both by examining their internal representations (embeddings), and by comparing their performance to that of humans. To assess the encoding of stimulus complexity by neural networks, we generated 1,000 stimuli using the LoT model, mirroring the approach of (Sablé-Meyer et al., 2022). We aimed to determine whether the embeddings in our models captured the Minimum Description Length (MDL) of the stimuli, used as a measure of their complexity. We first obtained embeddings for the stimuli and then conducted both decoding and correlative analyses. A linear regression model was trained to predict the MDL from the embeddings using 20-fold cross-validation and evaluate the extent to which model embeddings varied as a function of stimulus complexity. Additionally, we calculated the Euclidean distance between each stimulus embedding and the average embedding across all stimuli. The correlation between these distances and the MDL was computed to evaluate the alignment of our embedding metrics with stimulus complexity.

We found that embeddings taken from all three models contained robust information about stimulus MDL (Fig. 2a) with slightly higher decoding performance for CLIP and DINOv2 ($R^2 = 0.61$ and $R^2 = 0.65$, respectively) compared to ResNet-50 ($R^2 = 0.59$) as evaluated with 20 fold cross validation. Moreover, a simple correlation analysis found that the distance to the average embedding for each target stimulus was weakly, but significantly, correlated with the MDL for all three models ($p < 0.01$ for all).

We then compared metrics derived from these pre-trained neural networks on their ability to predict human performance using the same trials presented to human participants in the study by Sablé-Meyer et al. 2022 (Fig. 2c, significance reported in Table 1). For each trial, we extracted embeddings for the target and the five distractor stimuli from each network. Two key embedding metrics were computed to predict human performance. The first was the Euclidean distance between the target embedding and the centroid of the distractor embeddings, hypothesized to reflect the "confusability" of the target with its distractors. This metric might be related to the choice reaction times of participants. The second was the Euclidean distance between the target embedding and the centroid of the entire stimulus space, as introduced above. This may reflect the general "confusability" of the stimulus and potentially related to the time participants spent encoding each image.

We fit GLM models with the same confound variables used in Sablé-Meyer et al. 2022. For the choice condition, we fit one GLM for each neural network model with the target-distractor embedding distance metric derived from the neural networks and the confound variables as predictors

Table 1: DMTS Regression Significance ($p$-values)

| Condition | Regression | Model | Metric | p-value | $R^2$ |
|---|---|---|---|---|---|
| Choice | GLM | CLIP | target-distractor | 4.59e-06 | 0.91 |
| Choice | GLM | DINOv2 | target-distractor | 2.01e-07 | 0.93 |
| Choice | GLM | ResNet | target-distractor | 8.53e-05 | 0.89 |
| Choice | GLM | LoT | MDL | 3.73e-04 | 0.88 |
| Choice | Mixed Effects | CLIP | target-distractor | 9.72e-26 | 0.19 |
| Choice | Mixed Effects | DINOv2 | target-distractor | 2.11e-29 | 0.19 |
| Choice | Mixed Effects | ResNet | target-distractor | 1.77e-18 | 0.18 |
| Choice | Mixed Effects | Symbolic | MDL | 7.01e-03 | 0.18 |
| Encoding | GLM | CLIP | target-centroid | 7.58e-01 | 0.80 |
| Encoding | GLM | DINOv2 | target-centroid | 2.70e-02 | 0.84 |
| Encoding | GLM | ResNet | target-centroid | 3.42e-01 | 0.81 |
| Encoding | GLM | Symbolic | MDL | 1.19e-03 | 0.87 |
| Encoding | Mixed Effects | CLIP | target-centroid | 7.32e-01 | 0.11 |
| Encoding | Mixed Effects | DINOv2 | target-centroid | 1.43e-02 | 0.11 |
| Encoding | Mixed Effects | ResNet | target-centroid | 1.25e-01 | 0.11 |
| Encoding | Mixed Effects | Symbolic | MDL | 3.31e-03 | 0.11 |

in predicting the average human choice times for each stimulus. Likewise, we followed the same approach for the encoding times, but used the target-centroid distance metric from the neural network embeddings instead. We also fit GLMs with the same confounds and MDL as predictors to compare MDL with the embedding-derived distance metrics. This allowed us to compare the relative contribution of the metrics derived from the neural networks to the MDL in predicting human performance, by directly evaluating quality of model fits.

Due to concerns with model overfitting, we also fit linear mixed effects models at the trial level with target stimulus as a grouping variable to evaluate whether the same differences still held in this more challenging regression setting (Table 1).

In the regression analysis, GLM fits consistently indicated that the target-distractor distance metrics derived from the model embeddings predicted participant choice RT just as well or better than the MDL models (Table 1). Conversely, the DINOv2 target-centroid distance was the only network-derived metric that was significantly predictive of participant encoding time ($p < 0.05$), although the MDL model predicted human encoding RT better in this setting. Although the prediction of encoding reaction time from the target-centroid distance metric from DINOv2 was significant, the metric fails to capture the same shape of the trend that MDL has (in which the MDL increases as reaction time increases). Future work may involve searching for more appropriate metrics for encoding times that can be derived from DINOv2's embedding space. Linear mixed effects models recapitulated the general trend that network-derived distance metrics were just as predictive of human RT as MDL during the choice phase but not during the encoding phase. Notably, for choice RT modeling, the comparable performance of the network-derived metrics' to MDL in predicting choice RT was most evident for the larger models compared to ResNet, although further analysis is required to confirm this observation. Our results underscore that it is possible to predict behavior just as well as the symbolic model used in Sablé-Meyer et al. 2022 using simple metrics derived from the representations of large pretrained neural networks. We may conclude from this that human-like intuitions of subjective geometric complexity may already be contained in the representations of such large pretrained networks without the need for symbolic structure.

## 4 GEOMETRIC REGULARITY

### 4.1 BACKGROUND

The Quadrilateral Oddball Task, was used in Sablé-Meyer et al. 2021 to compare the ability of diverse human groups—differing in education, cultural background, and age—to that of non-human primates, symbolic models, and neural networks in their sensitivity to geometric regularity. Par-

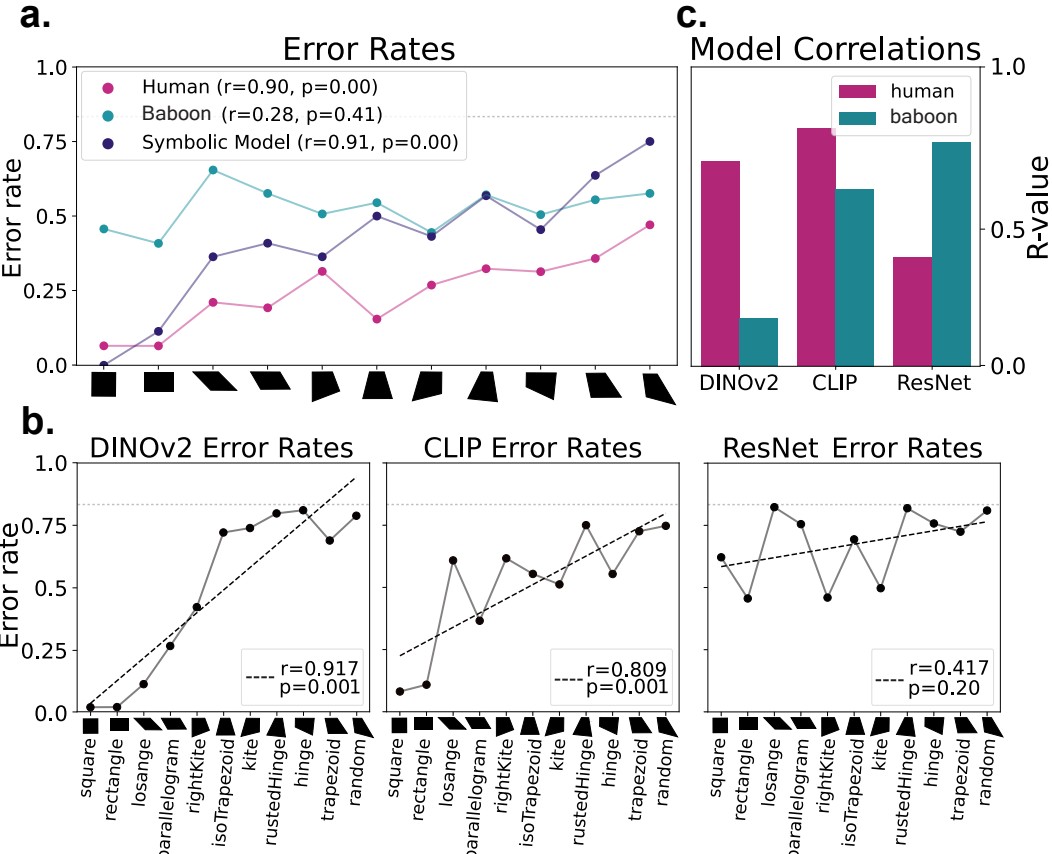

Figure 3: **Comparison of model biases to human/baboon performance on quadrilateral oddball task.** A) Human, baboon, and symbolic model error rates on the quadrilateral oddball task reported by Sable-Meyer et al (2021), sorted by shape geometric regularity (most regular on the left to least regular on the right). B) Neural network error rates as evaluated on the same trials shown to human and baboon participants in the original task. C) Bar plot displaying r-values for statistical tests evaluating correspondence of model error rates with human and baboon error rates.

ticipants were presented with a set of five reference quadrilaterals alongside a singular "oddball" quadrilateral, and tasked with identifying the oddball (see Fig. 1b). The reference quadrilaterals were constructed to vary in symmetry and regularity (such as number of parallel lines, congruent sides, and equal angles), yielding a range from perfect squares (the most regular quadrilateral) to random quadrilaterals (devoid of any such regularities). On each trial, participants were presented with five variants of a reference shape drawn from the set of quadrilaterals, that differed only in size and orientation, while the oddball was a perturbed version of the reference shape with the lower right vertex altered to disrupt any inherent regularity in the shape (Fig. 1b).

Sablé-Meyer et al. 2021 found that humans exhibited the highest accuracy in identifying the oddball when generated from (and presented among) highly regular shapes, with performance progressively declining as the reference shapes became more irregular (Fig. 3a). In contrast, non-human primates and simple neural network models (a ConvNet like the ResNet used in the present work) achieved scores above chance, but failed to demonstrate any discernible performance biases as a function of geometric regularity. Sablé-Meyer et al. 2021 showed that a symbolic model, built from an explicitly symbolic feature space derived from the discrete geometric properties of the shapes, could predict human performance in the task, and in particular the trend of increasing errors with greater geometric irregularity (Fig. 3a).

## 4.2 RESULTS

To assess the correspondence between representations in the large pre-trained neural networks and the biases exhibited in human behavior on this task, we presented the same set of images to the models that were shown to human and non-human primate participants in the original study, totaling approximately $60,000$ trials. On each trial, the six images of quadrilaterals used in the corresponding trial of the empirical study were presented to the model . For each image, the model's internal representations were extracted, and the outlier (as a proxy for the model's identification of the oddball) was determined by identifying the embedding that had the greatest Euclidean distance from the others (note that this is the same procedure Sablé-Meyer et al. 2021 use to evaluate the Oddball task on neural network models). We computed an average error rate for each reference shape (i.e., failure to identify the oddball embedding as the outlier in the display) across all trials, and ordered these error rates according to the geometric regularity of the reference shapes. The error rates from each model were then correlated with those of humans and non-human primates to compare the model's performance to that of each group.

The larger, self-supervised models DINOv2 and CLIP were better aligned with human performance biases in the geometric oddball task (Fig. 3bc), and exhibited a robust preference towards geometrically regular shapes, as evidenced by a significantly positive slope ($p < 0.001$) in their error rates as a function of regularity (Fig. 3). Moreover, although the error rates derived from the CLIP embeddings were significantly correlated with the baboon error rates ($p < 0.05$), both CLIP and DINOv2 error rates most closely matched human error ($p < 0.01$). Consistent with prior work (Sablé-Meyer et al., 2021), the smaller ResNet-50 model trained on classification more closely matched baboon error rates, and did not exhibit a significant positive trend as a function of geometric regularity.

## 5 GEOMETRIC PARTS & RELATIONS

### 5.1 BACKGROUND

The Geoclidean Task (Fig. 1c), introduced by Hsu et al. 2022, leverages a domain specific language (DSL) designed to encapsulate the fundamental elements of Euclidean geometry. This task uses the Geoclidean DSL to define and render geometric concepts, which in turn facilitates the investigation of geometric generalization capabilities. The Geoclidean DSL enables the generation of diverse images that embody the same abstract geometric concept through a set of Euclidean construction rules. Hsu et al. 2022 developed two datasets based on this system: Geoclidean-Elements, which draws from the axioms and propositions found in the first book of Euclid's *Elements*, and Geoclidean-Constraints, which offers a more streamlined examination of the relationships between Euclidean primitives (see Fig. 1c).

This task is designed to probe the intuitive understanding of Euclidean geometry, as evidenced by the ability to generalize from a limited set of examples to new instances. Hsu et al. 2022 demonstrated that humans show a robust capacity for such generalization across 37 different concepts, suggesting a natural sensitivity to the hierarchical structure and part relations inherent in Euclidean geometry. Conversely, they found that neural network models pretrained on ImageNet (which were more similar in size to the ResNet model used in the present work) struggle with few-shot generalization in this context.

### 5.2 RESULTS

We replicated the task conditions used by Hsu et al. 2022 with human participants as closely as possible in our evaluation neural network performance, in order to test the hypothesis that larger model size and enhanced training procedures would significantly reduce the gap between human and model performance.

Following Hsu et al. 2022, we assessed the proficiency of our three neural network models on the Geoclidean Task by testing each model's accuracy in making category judgments under "far" and "close" conditions, analogous to the task set presented to human participants in the original study.

DINOv2 and CLIP outperformed the ResNet model on this task; however, their accuracy (66% and 70% respectively; see Table 1) fell considerably short of human performance (91%). Successful per-

Table 2: Geoclidean Task Performance

| Model | Overall | Close | Far |
|-------|---------|-------|-----|
| CLIP | 0.70 | 0.67 | 0.73 |
| DINOv2 | 0.66 | 0.64 | 0.69 |
| ResNet | 0.64 | 0.61 | 0.67 |

formance of this task depends on the capacity to interpret the relationships and intersections among the sub-components of each stimulus that define each category. The complexity of the task is heightened by the necessity to identify not only the presence of specific sub-parts in each stimulus, but also to understand how they are arranged to create the overall Euclidean structure. This is particularly challenging as both category members and outliers are composed of identical components, with their relational configurations being the sole differentiating factor. Static embeddings derived from these pre-trained models may not be expressive enough to encode these relational features, and that may explain why their accuracy on this task lags so far behind human performance. In the Discussion below, we consider ways in which neural networks models might be augmented to address these challenges.

## 6 DISCUSSION

Geometry is a domain in which humans form useful abstractions that capture regularities across stimuli. A prevailing theory in cognitive science is that these geometric abstractions reflect symbolic structure in human mental representations (Fodor, 1975; Dehaene et al., 2022). Along similar lines, it has been argued that, without explicitly imbuing neural networks with symbol processing capabilities, they will not be able to exhibit the same cognitive flexibility as humans (Marcus, 2018; Fodor & Pylyshyn, 1988). Our results suggest that, contrary to this hypothesis, large neural networks pre-trained on massive numbers of images and text have the capability to reproduce key behavioral phenomena that have been associated with abstract geometric reasoning and symbol processing capability.

First, humans have an intrinsic notion of *geometric complexity* that influences their perceptual behavior. Sablé-Meyer et al. 2022 explored this using a working memory Delayed Match to Sample Task (DMTS) in which participants had to memorize a target image and select it among distractors after a 2 second delay period (Fig. 1a). Behavioral metrics, such as how long people had to make a decision (choice times), were predicted by a shape's Minimum Description Length (MDL), the length of the shortest symbolic drawing program needed to generate the shape (Fig. 2b). We show that the embedding distances in large pre-trained models such as CLIP and DINOv2 provide an equally or better predictor of human choice times (Fig. 2c). Further, these distances appear to correlate closely with MDL (Fig. 2a), suggesting that neural networks can learn representations that contain information about geometric complexity that humans are sensitive to, without explicitly being imbued with symbol processing capabilities.

Second, humans have a strong bias towards *geometric regularity* (Sablé-Meyer et al., 2021), with particular sensitivity to abstract features such as right angles, parallel lines, and symmetry. Sablé-Meyer et al. 2021 designed an Oddball task to probe this specific human bias, in which participants had to identify an Oddball shape out of a group of six quadrilaterals in which the Oddball violated a specific regularity in the group of quadrilaterals (Fig. 1b). We found that, consistent with findings of Sablé-Meyer et al. 2021, a ResNet model (a standard convolutional neural network architecture trained on object classification) failed to reproduce the geometric regularity effect, in which performance improved with the amount of geometric regularity in the trial stimuli (Fig. 3). At the same time, we showed that CLIP and DINOv2—transformer architectures trained on larger datasets using self-supervised objectives—*do* reproduce the geometric regularity bias (Fig. 3).

Third, humans are predisposed to parse geometric shapes into separable parts and relations among those. Hsu et al. 2022 built the Geoclidean task to demonstrate this capability in humans, and benchmark it in artificial intelligence systems. In the Geoclidean task, subjects have to make category judgements based on stimuli that were generated using a DSL that hierarchically generates geometric visual stimuli with multiple parts and relations (Fig. 1c). From just a few exemplars,

participants have to learn the category embodied by the set of exemplars, and then generalize this to identify whether a novel stimulus is a member of the category. We found that DINOv2 and CLIP outperform ResNet on the Geoclidean task, but fall short of human-level performance. This suggests further work is needed to endow neural networks with the human-like ability of decomposing an abstract geometric stimulus into parts and relations. However, whether or not this requires the explicit addition of symbolic processing capabilities remains an open question. Recent work has shown that object-centric representations can be learned in visual transformers using Slot Attention (Locatello et al., 2020), or in modifying transformer architectures to be more sensitive to relations between objects than individual object features (Altabaa et al., 2023). These architectural augmentations do not specifically implement symbolic processing capabilities, though they may serve as inductive biases that may pressure neural networks to acquire such capabilities through learning (Webb et al., 2023a). Incorporating these kinds of methods at scale may be a path forward in enabling a human-like bias towards geometric parts and relations in neural network architectures.

Recent advancements in machine learning have revealed limitations in advanced transformer models, including GPT-4 (Achiam et al., 2023) and Gemini Ultra (Team et al., 2023), particularly in tasks requiring multi-step reasoning. Geometric reasoning tasks, which often demand sensitivity to relations among multiple object sub-parts (as in the Geoclidean constraints task) or require the comparison of several stimuli across different dimensions (as seen in matrix reasoning tasks), have posed similar difficulties for these models. These shortcomings can be mitigated by introducing an explicit sequential processing mechanism, a strategy exemplified by techniques like chain-of-thought (CoT) and tree-of-thought (ToT) (Yao et al., 2023; Prystawski & Goodman, 2023), where the model performance can be improved by explicitly being told to give intermediate reasoning traces leading to the final answer (although there are occasions when these reasoning traces are not faithful representations to the actual reasoning behind a model's prediction, Turpin et al. 2023). Exploring the application of serialization methods, similar to ToT and CoT, could represent a promising direction for enhancing machine performance to more closely approximate human capabilities in geometric reasoning tasks.

Across our experiments, we saw a general trend that DINOv2 outperformed CLIP in reproducing human behavior. Both models have a similar architecture (i.e. transformer-based) and were trained on comparable amounts of data. The main difference between these two models is that CLIP was trained jointly on vision and language whereas DINOv2 was only trained on images. It is possible that jointly training models with language along with vision does not necessarily aid in developing geometric visual abstractions, which corroborates neuroimaging work that suggests the brain networks for mathematical reasoning and language processing are separate (Amalric & Dehaene, 2018; Ivanova et al., 2020).

Our work shows that DINOv2 and CLIP, which are transformer architectures pre-trained on large-scale data, produce more human-like patterns of geometric processing than ResNet, a standard convolutional neural network architecture trained on ImageNet. DINOv2 and CLIP differ from ResNet in model architecture, scale of data distribution, model size, and training objective. Each of these differences may be a factor contributing to our results. There is work showing that, like humans, transformer architectures are biased more towards shapes whereas convolutional neural networks are biased more towards texture (Tuli et al., 2021). Additionally, work in machine learning has found that scaling up model and training data size can change model capabilities (Wei et al., 2022). Self-supervised contrastive objectives have also been shown to lead to more complex visual reasoning (Ding et al., 2021). Future work will entail disentangling the importance of each of these factors in leading to human-like visual geometric processing.

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
