# OpenReview forum: "Human-like Geometric Abstraction in Large Pre-Trained Neural Networks"
_ICLR.cc/2024/Workshop/Re-Align — ICLR 2024 Workshop Re-Align Poster_

### Official Review · Reviewer_BUjU · 2024-02-19
**Interesting paper with well-supported experiments**

**Rating:** 3
**Fit:** 3
**Confidence:** 2

**Workshop Review:**

The authors investigated this question: do deep learning models identify geometric shapes similar to human beings? To answer this question, they studied the dependence of language models on three features: geometric complexity, geometric regularity and parts & relations, and compare the results to humans. They found that language-vision models are more like human performance than pure vision models. This is an interesting finding, which may have implications for both machine learning and cognitive science.

This paper is well written and well motivated. Conclusions are well supported by experiments. The only concern I have is lack of literature review. The paper would be further improved if a related work section is included. Also, it is probably out of the scope of the paper, but I would love to see more implications/evidence of how these results imply how humans perceive and reason about geometric shapes.

**Reason For Not Giving Higher Score:**

No higher score.

**Reason For Not Giving Lower Score:**

The paper is solid, and well-written. A pleasure to read.

**Reviewer Domain:**

machine learning

---

### Official Review · Reviewer_oYKq · 2024-02-26
**An creative evaluation study comparing human behavior and artificial neural networks and vision foundation models**

**Rating:** 3
**Fit:** 3
**Confidence:** 3

**Workshop Review:**

This works compares human behavior and pretrained vision-language models (CLIP and DINOv2) on a task assessing 3 key behaviors in visual cognition: a sensitivity towards complexity, a preference for regularity, and the perception of parts and relations.
A general comment is to split figure 1 into 3 subfigures and put the corresponding visualizations of the task with the results to better contextualize them.
In the complexity section, I found the result that the model embeddings encode information about the MDL of the stimuli very fascinating. Could the authors explore this more mechanistically? Is the dimensionality of the representations different for different MDL stimuli? What is the source of the information about MDL in these representations? How is this metric changing as a function of depth in the transformer?
The geometric regularity and geometric parts and relations results were less convincing. For geometric regularity, the main finding is that the error rates are correlated with geometric regularity of the shape, for CLIP and DINOv2. Again, could the authors study this more mechanistically? What features is the model relying on when making the same errors as humans? Since these are transformer models, it is easy to do basic feature attribution or attention-map visualization. This will make the results more illuminating.
For parts and relations, I find the results the weakest. The R^2 table is all that is shown and although there seems to be a trend from Resnet-> DINOv2, the change in R^2 value is small. Another statistical metric to compare goodness of fit across these models is needed. The authors have acknowledged that all models fall short of human performance and thus it is a useful contribution to understand the limitations of current state of the art models. This task is also harder and thus some mechanistic analyses would tell the reader more about WHY the models are failing to reach human-level performance at this task. Also, another study the authors could perform is fine-tuning. The model might contain information relevant to the task but this might be suppressed by the model outputs. Fine-tuning could unlock this capability. It would be a great result if this is true.

Another general comment is the lack of any feature attribution or visualization. Specially DINOv2 was designed for simple feature visualization and thus I would urge the authors to add a section on it for all 3 tasks considered, for a future version of the paper. Also the TLDR seems to focus on a previous result which was about symbolic reasoning whereas the overall results seem very relevant broadly for deep learning in general.

**Reason For Not Giving Higher Score:**

The lack of any mechanistic analyses about the purely behavioral results. Specially relevant since for 2 out of 3 tasks, the best models match human biases while for the third, the models fail. Why?

**Reason For Not Giving Lower Score:**

N/A

**Reviewer Domain:**

neuroscience

---

### Official Review · Reviewer_nD4V · 2024-02-28
**Human-like Geometric Abstraction in Large Pre-Trained Neural Networks**

**Rating:** 2
**Fit:** 3
**Confidence:** 2

**Workshop Review:**

**Summary**

The manuscript evaluates the ability of two self-supervised transformer-based NN models (DINOv2 and CLIP) and a standard CNN (ResNet-50) to reproduce correlates of human and primate behaviour in geometric visual processing tasks. These tasks are meant to evaluate key features of visual processing, including sensitivity towards complexity, preference for regularity and symmetry, and geometric edge and part relations. These results challenge the school of thought in cognitive science that necessitate models that incorporate explicit symbols and discrete categories for higher-order geometric reasoning.

*Experiment 1*: Sable-Meyer et al., 2022 measured human reaction times during the encoding and choice phases of a delayed match-to-sample task and found that these reaction times were linearly related to a measure of the length of the program used to generate these images, called the Minimum Description Length (MDL). The authors use two measures: (1) distance between correct vs. centroid of distractor stimuli, and (2) distance between correct vs. centroid of stimulus space, to fit the encoding and choice reaction times respectively. While the choice time was fit well, the encoding time was fit poorly.

*Experiment 2*: Sable-Meyer et al., 2021 measured the ability of humans, baboons and a symbolic model to identify an ‘oddball’ stimulus among a set that had a specific asymmetry. The error rate for all three agents increased with geometric irregularity of the base shape. The error rates of DINOv2 and CLIP were similar to that of humans, whereas ResNet-50 had error rates that were higher and more similar to the nonhuman primate data.

*Experiment 3*: Hsu et al., 2022 asked participants to identify if a target image is within, close or far from a set of images, based on Edge relations or Part relations. While DINOv2 and CLIP outperformed ResNet-50 on this task, they fell short of human performance.
In general, the performance of DINOv2 > CLIP > ResNet-50. This is interesting since CLIP is trained on both images and language, whereas DINOv2 is only trained on images. The work shows that complex NNs can implicitly model geometric complexity, regularity and relational information without explicitly coded symbols.

I like that the authors use different studies that examine various aspects of human and nonhuman primate geometric visual reasoning. The results are well-described and discussion is comprehensive.

**Suggested edits and comments:**

*Experiment 1*: “We fit GLM models with the same confound variables used in Sable-Meyer et al. 2022.” I do not find this to be a sufficient description. You should clearly explain what the independent and dependent variables are in each of the models that you describe.

In the original work, a single variable (MDL) is shown to be linearly related to both choice times and encoding times. It is unclear to me why the authors chose to use two different variables to fit these. This will only work if the two independent variables (target-distractor distance and target-stim space distance) are linearly related to each other, which does not seem likely. I would suggest only using the target-distractor distance, since that makes the central point of the paper, i.e. to find a NN parameter that fits the human behaviour.

*Figure 2*: Please add $R^2$ and p-values for the fits shown in subpanel c.

*Figure 3*: Please do not say $p=0.00$ – show the correct value using scientific notation, like you do in Table 1. In this figure, R-values are shown whereas $R^2$ values are used in Fig. 2 and Table 1. Please be consistent.

*Experiment 3*: I find the description of the original experiment to be lacking clarity. The authors describe the approach of the Geoclidean DSL in too much detail, in my opinion. It is less clear how the task is structured. I suggest including more details about the actual experiment instead.

**Reason For Not Giving Higher Score:**

I believe the work is good but exploratory and needs more polish to be of sufficient impact warranting a talk. While the authors show that there are parameters in complex transformer-based NNs that can correlate to human performance, a general principle of discovering such parameters is not discussed or provided.

**Reason For Not Giving Lower Score:**

N/A

**Reviewer Domain:**

neuroscience

---

### Decision · Program_Chairs · 2024-03-02

Accept (Poster)